# Relationship between Perceived Training Load, Well-Being Indices, Recovery State and Physical Enjoyment during Judo-Specific Training

**DOI:** 10.3390/ijerph17207400

**Published:** 2020-10-11

**Authors:** Ibrahim Ouergui, Emerson Franchini, Okba Selmi, Danielle Evé Levitt, Hamdi Chtourou, Ezdine Bouhlel, Luca Paolo Ardigò

**Affiliations:** 1High Institute of Sports and Physical Education of Kef, University of Jendouba, Boulifa University Campus, Kef 7100, Tunisia; ouergui.brahim@yahoo.fr (I.O.); okbaselmii@yahoo.fr (O.S.); 2Martial Arts and Combat Sports Research Group, School of Physical Education and Sport, University of São Paulo, 05508-030 São Paulo, Brazil; emersonfranchini@hotmail.com; 3Applied Physiology Laboratory, Department of Kinesiology, Health Promotion, and Recreation, University of North Texas, Denton, TX 76203, USA; dlevit@lsuhsc.edu; 4Department of Physiology, School of Medicine, Louisiana State University Health Sciences Center New Orleans, New Orleans, LA 70112, USA; 5High Institute of Sport and Physical Education of Sfax, Université de Sfax, Sfax 3000, Tunisie; h_chtourou@yahoo.fr; 6Physical Activity, Sport and Health, UR18JS01, Observatoire National du Sport, Tunis 1003, Tunisie; 7Laboratory of Cardio-Circulatory, Respiratory, Metabolic and Hormonal Adaptations to Muscular Exercise, Faculty of Medicine Ibn El Jazzar, University of Sousse, Sousse 4000, Tunisia; ezdine_sport@yahoo.fr; 8Department of Neurosciences, Biomedicine and Movement Sciences, School of Exercise and Sport Science, University of Verona, 37131 Verona, Italy

**Keywords:** combat sports, training load monitoring, randori, uchi-komi, martial arts

## Abstract

This study investigated the relationship between well-being indices and the session rating of perceived exertion (session-RPE), recovery (TQR), and physical enjoyment (PE) during intensified, tapering phases of judo training. Sixty-one judo athletes (37 males, ranges 14–17 years, 159–172 cm, 51–67 kg) were randomly assigned to three experimental (i.e., randori, uchi-komi, running) and control groups (regular training). Experimental groups trained four times per week for 4 weeks of intensified training followed by 12 days of tapering. Session-RPE, well-being indices (i.e., sleep, stress, fatigue, delayed onset of muscle soreness (DOMS), Hooper index (HI)), and TQR were measured every session, whereas PE was recorded after intensified, tapering periods. Recovery (TQR) was negatively correlated with sleep, stress, fatigue, DOMS, HI, session-RPE in intensified period and was negatively correlated with sleep, stress, fatigue, DOMS, HI in tapering. Session-RPE was positively correlated with sleep, fatigue, DOMS, HI in intensified period and positively correlated with fatigue, DOMS in tapering. PE was negatively correlated with stress in intensified training. Enjoyment could be partially predicted by sleep only in intensified periods. Session-RPE could be partially predicted by TQR, fatigue during intensified periods and by sleep, and HI during tapering. Sleep, recovery state, pre-fatigue states, and HI are signals contributing to the enjoyment and internal intensity variability during training. Coaches can use these simple tools to monitor judo training.

## 1. Introduction

Reaching peak judo performance requires the optimal development of physical, physiological and psychological abilities. To achieve the desired profile, several judo-specific training methods have been developed and compared [1,2,3]. To improve athletes’ physical and physiological performances, high-intensity interval training (HIIT), including general and specific training methods, has been used in judo [2]. Sport-specific exercises have the potential to develop physical outcomes and physiological responses as well as technical and tactical aspects of the modality [2]. Regularly, combat simulation (randori) or throwing technique exercises (uchi-komi, technique repetition without throwing the partner and nage-komi, throwing technique repetition throwing the partner) as specific training methods are used in judo [4]. High-intensity interval training using uchi-komi as exercise and performed in all-out mode improves anaerobic and aerobic fitness [2] while allowing technical development and guaranteeing that specificity is provided [4].

Careful monitoring of training load, recovery and changes in psychological status are important for training periodization, to improve athletic performance and to avoid problems related to non-functional overreaching [5,6]. Considering and predicting these variables can aid in planning and refining training programs and could potentially help to avoid injury and burnout. In fact, Branco et al. [6] reported that there were no differences in the rating of perceived exertion (RPE) and internal training load over 4 weeks of HIIT added to regular judo training when comparing uchi-komi and upper- and lower-body cycle ergometer exercises. Furthermore, possible interactions between these variables may occur during a single exercise session or training period and may affect athletic performance. For example, the relationships between well-being indices, recovery state, RPE and athletic performances (i.e., technical and physiological aspects) have been established in team sports [7,8]. Selmi et al. [8] reported that fatigue and training load increase and recovery insufficiency during training sessions affected sports performance, induced psychological disturbance and increased the risk of injuries. Furthermore, a single study in judo athletes reported that an improved mood and decreased muscle soreness preceded increased performance during a tapering period following 2 weeks of intensified training in judo athletes [9], demonstrating the importance of monitoring these variables among combat sports athletes. Because of the potential impacts on performance and risk of injury, it has been largely recommended to use subjective indices to estimate exertion, wellness and recovery state as common methods for monitoring psycho-physiological responses in athletes, especially during intensified training periods.

To the authors’ current knowledge, studies in combat sports that have monitored internal training load and recovery state during extended intensified training followed by tapering periods are lacking. Over tapering periods, by definition, there is a training load decrease and this might variously influence internal training load, recovery state, physical enjoyment and well-being indices. Relationships between internal training load and recovery state have not been investigated in combat sports over a whole training period (i.e., intensified training + tapering). Thus, the objective of the present study was to investigate the relationship between well-being indices (i.e., sleep, stress, fatigue, delayed onset muscle soreness (DOMS), Hooper index (HI)) and total recovery state (TQR)), internal training load (session-RPE) and physical enjoyment over intensified and tapering periods. It was hypothesized that physical enjoyment and internal training load are related to and predicted by well-being indices and recovery state.

## 2. Materials and Methods

A similar subject population and general methods were previously used for another study [10] and this complementary analysis was conducted to better understand the relationship between internal training load and recovery over different training phases. Using the G*Power software (Heinrich-Heine-Universität, Düsseldorf, Germany) with α and power fixed at 0.05 and 0.95, respectively and effect size fixed at 0.3 based on the study of Franchini et al. [2], the sample size was calculated a priori. The required sample size was 40, but a higher number was recruited considering the possibility of dropouts during the training process. Sixty-one judo athletes (37 males) volunteered to participate in the study. Athletes were randomly assigned to randori (i.e., grip dispute practice without throwing technique, RG, 8 males and 6 females, age 16 ± 1 years, height 171.0 ± 0.8 cm and body mass 58.4 ± 7.4 kg [mean ± SD]), uchi-komi (i.e., technique repetition training, UG, 10 males and 6 females, age 15 ± 1 years, height 164.2 ± 0.8 cm and body mass 58.9 ± 8.3 kg), running (RuG, 11 males and 8 females, age 15 ± 1 yrs, height 162.5 ± 1.1 cm and body mass 55.8 ± 3.2 kg) and control group (CG, 6 males and 6 females, age 15 ± 1 years, height 159.2 ± 0.5 cm and body mass 56.8 ± 5.4 kg). The three different training programs were chosen as three different modalities of combat simulation (randori), combat fundamentals learning (uchi-komi) and general athletic conditioning (running) to investigate their specific effect on internal training load, recovery state, physical enjoyment and well-being indices. They trained four sessions per week (1 h 30 min/session) and had more than 7 yrs of experience in judo practice with regular participation in competitions during the prior two years. Athletes did not present any medical restrictions and were not in the process of decreasing body mass during the experimental period. This study was conducted according to the Declaration of Helsinki for human experimentation [11] and the protocol was fully approved by the local research ethics committee before its beginning. All athletes and their parents gave written informed consent after a detailed explanation about aims and risks involved in the investigation.

After being well familiarized with all scales and questionnaires, in addition to their usual technical-tactical training, athletes participated in supplementary HIIT—except participants in the control group—over 4 weeks with two sessions per week followed by twelve days of tapering. Before each training session, athletes were submitted to 15 min of standardized warm-up consisting of jogging, dynamic stretching and technique skills practice.

During these two periods, TQR scale from 6 to 20 (ranging from “very very poor recovery”, 6 points, to “very very good recovery”, 20 points) and well-being indices scores using a scale from 1 to 7 points (i.e., stress, sleep quality, fatigue level, DOMS and sum of these four variables used to calculate HI [12]) were collected 15 min before each training session. Delayed onset muscle soreness was used to rate the overall body soreness as previously reported by Papacosta et al. [9]. Well-being indices showed good reliability in terms of Intraclass Correlation Coefficient (ICC) ranging from 0.60 to 0.81 in the present study. The TQR scale was used to assess athletes’ recovery state and was reflective of response to preceding training day [13]. The TQR scale showed an ICC value of 0.823 in the present study.

Rating of perceived exertion (RPE, Borg’s CR-10 scale [14]) was recorded for each athlete 15–30 min after each training session and perceived training load (i.e., session-RPE) following each session for each judo athlete was calculated from the product of the session duration and the athlete’s RPE [14]. The Intraclass Correlation Coefficient for RPE was 0.808. Mean values for these variables were calculated for both training periods and used for analysis. Moreover, physical enjoyment calculated from the sum of 18 items of the Physical Activity Enjoyment Scale (PACES [15]) was assessed after both intensified and tapering training periods. Physical Activity Enjoyment Scale showed an ICC value of 0.87 in the present study. All training sessions and measures were conducted at the same time of day (5:00 p.m.–7:00 p.m. [16]) during the competition period of the season.

Exclusive of warm-up, each HIIT session duration was 14 min 40 s and included two blocks of 4 min 50 s of high-intensity intermittent exercises (10 × 20 s effort with 10-s pause) separated by 5 min of recovery [2]. All training modalities were executed in an all-out mode [2]. For the randori group, athletes were selected according to weight categories and were asked to spar in a standing combat phase ensuring kumi-kata (grip dispute) and executing techniques without throwing. For the uchi-komi group, athletes were asked to perform repetitive technical entrance without throwing with a single partner from the same weight category. Athletes in the running group performed running back and forth as fast as possible between two lines separated by 6 m as recently reported [10].

Data are presented as mean and standard deviation. Statistical analysis was performed using SPSS 20.0 statistical software (SPSS Inc, Chicago, IL, USA). The normality of datasets was checked and confirmed using the Kolmogorov–Smirnov test. Pearson’s product-moment correlation coefficient was used to examine relationships between self-subjective ratings of sleep quality, fatigue, stress, DOMS and TQR with RPE and physical enjoyment. Magnitude of correlation was determined as trivial when *r* < 0.1, low 0.1–0.3, moderate 0.3–0.5, large 0.5–0.7, very large 0.7–0.9, nearly perfect > 0.9 or perfect = 1 [17]. Multiple regression analyses were used to establish the relationship between the dependent variables (session-RPE and enjoyment) and the independent variables (TQR, sleep, stress, fatigue, DOMS and HI). After collinearity analysis, multiple linear regressions were determined for each training period (i.e., intensified and tapering phases) across all groups. The correlation coefficient value (*r*) was determined and the coefficient of determination (*r^2^*) was calculated to determine the proportion of the variability in the independent variables explained by the dependent variables. The standard error of estimate (SEE) of each equation is also presented. Statistical significance was set at *p* < 0.05.

## 3. Results

Results of correlational analyses (i.e., correlation coefficients and significances) between well-being indices, TQR, RPE and physical enjoyment measured over intensified and tapering periods for different judo training groups are presented in Appendix A, respectively.Most relevant correlations are the following. During intensified training period, TQR was negatively correlated with sleep (*r* = −0.5, *p* < 0.01, *very large*), stress (*r* = −0.35, *p* < 0.01, *moderate*), fatigue (*r* = −0.42, *p* < 0.01, *moderate*), DOMS (*r* = −0.35, *p* < 0.01, *moderate*), HI (*r* = −0.49, *p* < 0.01, *moderate*) and RPE (*r* = −0.62, *p* < 0.01, *large*). Moreover, RPE was positively correlated with sleep (*r* = 0.29, *p* < 0.05, *low*), fatigue (*r* = 0.40, *p* < 0.01, *moderate*), DOMS (*r* = 0.41, *p* < 0.01, *moderate*) and HI (*r* = 0.37, *p* < 0.01, *moderate*). Finally, physical enjoyment was negatively correlated with stress (*r* = −0.25, *p* < 0.05, *low*).

During tapering period, TQR was negatively correlated with sleep (*r* = −0.52, *p* < 0.01, *large*), stress (*r* = −0.68, *p* < 0.01, *large*), fatigue (*r* = −0.71, *p* < 0.01, *very large*), DOMS (*r* = −0.60, *p* < 0.01, *large*), HI (*r* = −0.77, *p* < 0.01, *very large*). Rating of perceived exertion was positively correlated with fatigue (*r* = 0.30, *p* < 0.05, *moderate*), DOMS (*r* = 0.26, *p* < 0.05, *low*).

Regression analyses showed that enjoyment could be partially predicted from sleep during the intensified period using the following equation:Enjoyment (a.u.) = 76.61 + 3.514 (sleep in a.u.), *r* = 0.259, *r^2^* = 0.067, SEE = 6.497 and *p* = 0.04.(1)

Enjoyment could not be predicted in the tapering phase (*p* > 0.05). Session-RPE could be partially predicted in the intensified and tapering phases using the following equations:intensified phase: session-RPE (a.u.) = 1109.21 − 42.86 (TQR in a.u.), *r* = 0.622, *r^2^* = 0.387, SEE = 67.75 and *p* < 0.001 and(2)
tapering phase: session-RPE (a.u.) = 137.05 − 18.64 (sleep in a.u.) + 9.247 (HI in a.u.), *r* = 0.460, *r^2^* = 0.212, SEE = 25.82 and *p* = 0.001.

## 4. Discussion

The present study aimed at investigating relationships between well-being indices, TQR, RPE and physical enjoyment during intensified and tapering periods for different judo training modalities. Session-RPE and recovery were related to well-being indices during both training phases and physical enjoyment was inversely related to stress during the intensified training period (Appendix A. Only a small proportion (7–39%) of enjoyment and session-RPE could be predicted by the other variables. Therefore, the study’s hypothesis was partially confirmed.

Recovery state, sleep, stress, fatigue and DOMS may be used to control the prescription of training stimuli and prevent non-functional overreaching, negative affect toward training and deterioration in psycho-physiological processing [8]. It was previously found that well-being indices (i.e., sleep, stress, fatigue, DOMS and HI) and TQR were negatively correlated with each other during both intensified and tapering periods indicating a strong association between these psychometric variables and confirming the effectiveness of these tools in assessing states of athletes in any training periods [8]. In the present study, sleep disturbance, increased fatigue, stress and DOMS were inversely related to the recovery state of athletes during both training periods. Similarly, Selmi et al. [8] reported that four weeks of intensified training reduced recovery state and perceived recovery and were related to sleep quality, stress, fatigue and DOMS in professional soccer players. Thus, the results of the present study support the effectiveness of well-being indices (i.e., Hooper Index and subscales) and recovery state to identify stress after intense training [13,18] and their suitability for monitoring training loads representing a crucial factor to avoid potential maladaptive physiological effects of fatigue [19,20] However, given the small amount of variance in internal training load and enjoyment explained by well-being and recovery variables, more research regarding the relationships between these variables is needed. Whereas sleep, stress and fatigue were similarly related to TQR across experimental groups, DOMS was very largely correlated with TQR specifically in the randori group during intensified training. This result can be explained by the fact that these two training modalities solicited a larger amount of muscle mass and required more eccentric overload compared to the uchi-komi and running groups [21] and, in association with accumulated fatigue and stress due to training load increase, may have more profoundly impacted athletes’ recovery state.

Moreover, the present study showed correlations between TQR and well-being indices with session-RPE during the intensified training period. This result suggests that psychometric state and recovery may contribute to internal intensity signals especially during intensified training. Specifically, TQR and session-RPE are negatively related only in the intensified period with a large magnitude of correlation. This finding confirms the relationship between increased training load expressed by values of session-RPE and recovery process of athletes. Whereas TQR was largely and negatively related to session-RPE during the intensified training period, the relationship was absent during the tapering period likely as a result of improved recovery with lower training loads. In support of this finding, the beneficial physiological improvement due to tapering among judo athletes has previously been reported by Papacosta et al. [9]. Similarly, session-RPE was positively correlated with well-being indices. During the intensified training period, the magnitude of correlations with session-RPE was low with sleep and stress and moderate with fatigue, DOMS and HI. During the tapering period, the magnitude of correlation between session-RPE and fatigue and DOMS was low. The present study did not consider the coaches’ role during tapering. In reality, coaches could influence athletes’ recovery perception, i.e., conditioning them to focus on their strengths and disregard their weaknesses. This could explain found low correlation between session-RPE and fatigue and DOMS. The present study’s results are supported by Nédélec et al. [22], who reported that the training load was more strongly related to fatigue and muscle soreness than sleep and stress. The relationship between session-RPE and well-being indices was more pronounced during the intensified training versus the tapering period supporting that the reduction in training volume and frequency enhanced overall psychometric state after a period of intensified training. Session-RPE during the tapering period was explained by sleep quality and HI in the present study. Similarly, it has previously been reported that HI was associated with training load in professional soccer players [20] and reflected the physiological recovery time-course in athletes [23]. Moreover, it was reported that HI and subscales (e.g., sleep quality) are sensitive to training load fluctuations making them suitable for monitoring athletes [24]. Together with the results of the present study, previous findings support that sleep and psychological well-being mediate the effectiveness of the tapering period in promoting physiological recovery underscoring the importance of monitoring these variables in athletes.

However, not all studies have observed such relationships between subjective intensity and psychological well-being. For example, the results of the present study are not in agreement with the findings of Selmi et al. [25], who reported the absence of a relationship between well-being indices recorded before small-sided game training sessions and RPE recorded immediately after training sessions. Similarly, Haddad et al. [26] reported that RPE recorded after 10 min of the submaximal exercise was not influenced by the variability of well-being indices. Osiecki et al. [27] indicated no significant association between TQR and RPE measured 30 min after a match in male athletes involved in an official professional soccer match. These differences may be explained by the fact that the present study examined the relationship between internal training load, which accounts for session duration, and well-being indices across a whole training period, whereas the other studies investigated this relationship over single exercise bouts. In fact, the results of Moalla et al. [20] emphasized the need to investigate relationships between RPE and well-being indices also in combat sports athletes, because chronic effects may differ from acute responses. Taken together, the contrasting results of the present and previous studies underscore such acute and chronic differences. Nonetheless, the results of the present study confirmed that recovery quality and well-being indices reflected session-RPE [20,28] particularly during intensified training. Well-being variables measured before and session-RPE measured after judo training sessions were important for monitoring psychometric state of athletes during both intensified and tapering periods.

Finally, physical enjoyment is of great importance in assessing positive psychological responses to physical training. Indeed, physical enjoyment has been associated with positive responses to activity and with greater athlete motivation during training sessions [29]. The present study showed that physical enjoyment was negatively correlated with stress during the intensified training period, although sleep quality explained a small but significant amount of variability in physical enjoyment. These results may be explained by the negative psychological impacts of accumulated training stress during intensified training [30]. In fact, intense training may be related to increased mental fatigue and negative emotional states during exercise among athletes [31]. In contrast, Selmi et al. [25] reported that physical enjoyment was not related to TQR and well-being indices during an intense training cycle in soccer. Considering each training modality separately, physical enjoyment was largely unrelated to well-being indices, RPE, and TQR for all groups during both training periods suggesting that these variables minimally contribute to altered physical enjoyment during judo training modalities. Rather, physical enjoyment may be more closely related to positive emotional reactions and motivation during training [15,32,33,34,35]. The nuances of physical enjoyment during different training phases among combat sports athletes warrant further investigation.

A limitation of the present study is that the biological age of participants was not controlled. Despite being similar in chronological age (14–17 years), they were still in the process of growth and maturation and that could have influenced their internal training load, recovery state, physical enjoyment and well-being indices measures.

The highest coefficient of determination was 38.7% (Appendix A) suggesting that only a small part of the variables could be explained by other variables. In the present study, this result may be associated with the fact that these relationships could be influenced by other variables such as physical fitness, training experience, sex, weight category and genetic endowment. Another possibility is that the constructs of each variable are quite independent and therefore could not be predicted by other variables.

## 5. Conclusions

Overall, the present study showed that well-being indices (i.e., sleep, stress, fatigue, DOMS and HI) and TQR were negatively correlated during intensified and tapering periods. Moreover, RPE was positively correlated with well-being indices and negatively related to TQR during intensified and tapering periods. Recovery quality and fatigue explained a significant proportion of the variance in session-RPE during the intensified and tapering periods, respectively. These results demonstrate that recovery state and pre-fatigue states may contribute to internal intensity during both intensified and tapering periods among judo athletes. Moreover, physical enjoyment was minimally influenced by these variables. Thus, coaches can use simple and low-cost monitoring tools such as the well-being indices sleep and fatigue (particularly before training), total recovery state (TQR) scale and Hooper index (HI) during both intensified and tapering periods to control training stimuli and to adjust training regimens when necessary to support positive psychological states and properly-timed physiological recovery among judo athletes.

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
