# Peer review of "Relationship between Perceived Training Load, Well-Being Indices, Recovery State and Physical Enjoyment during Judo-Specific Training"

_ijerph, 2020, doi:10.3390/ijerph17207400_

Round 1
Reviewer 1 Report
Introduction
The introduction is clear but authors should develop whether, and why, they would expect different correlations between markers of psychological states during “normal” training and the tapering phase.
In addition authors should propose hypothesis to support the use of different experimental groups (randori, uchi-komi and running) and how these different modalities would influence perception of athletes.
Materials and methods
L 89: The number of male athletes in the present study differs from the other article using the same sample of participants, please correct.
L 102-103: The authors indicate that athletes from the different experimental groups, but not control group, participated to a supplementary HIIT training in addition to their usual training. The use of “supplementary” would suggest that experimental groups performed an additional training, and that the HIIT training did not replace part of the usual training. Therefore, the total training volume would be lower for the control group compared to the experimental groups, which could influence perceived responses to training. Could the authors rephrase this sentence to avoid misleading whenever the HIIT training replace part of the conventional training, otherwise, authors should provide the total training volume for the experimental and control groups and account for this difference in their discussion.
L 108 & 112: please provide the scales used to monitor RPE or DOMS, and details how DOMS were assessed and for which muscles groups.
L 111 : Could the authors provide references to support the methodology used to quantify session RPE? Indeed, an accurate monitoring of RPE should be conducted during exercise, or immediately at exercise completion (Pageaux B., 2016, European J of Sport Science)..
Results
The sole use of table in the results section is not well appropriate in this manuscript and somewhat boring to analyse. Authors should emphasize on main differences in the text, or the point of more interest regarding their analysis and discussion. I suggest therefore to reconsider the use of tables 1 and 2 by reducing the variables reported in the tables and clearly indicate in the text the most significant and relevant results.
Discussion
L 195-197: In addition to the lower training volume, authors should also consider the psychological state of the athlete during a training period. Specifically, athletes may optimise themselves their positive feelings during this period, and coaches could also play a determinant role when conditioning athletes for competitions by focusing on their strengths and avoiding they weakness. Howe athletes perceived their recovery state may thus be biased, and would therefore not relate accurately their physiological recovery level, which could partly supports the low correlation between session RPE and DOMS mentioned L 201-202.
Author Response
Response to Reviewer 1 Comments
Are the methods adequately described?
(x) Can be improved
Please, read below comments to specific points.
Are the results clearly presented?
(x) Can be improved
Please, read below comments to specific points.
Point 1: The introduction is clear, but authors should develop whether, and why, they would expect different correlations between markers of psychological states during “normal” training and the tapering phase.
Response 1: We thank expert reviewer for his/her suggestion. Whole aim paragraph was reworked as follows:
“Over tapering periods, by definition, there is a training load decrease and this might variously influence internal training load, recovery state, physical enjoyment and well-being indices. Relationships between internal training load and recovery state have not been investigated in combat sports over a whole training period (i.e., intensified training+tapering).”
Point 2: In addition, authors should propose hypothesis to support the use of different experimental groups (randori, uchi-komi and running) and how these different modalities would influence perception of athletes.
Response 2: Hypothesis to support use of different training modalities to investigate their effect on perception of athletes was specified in 2. Materials and Methods as follows:
“The three different training programs were chosen as three different modalities of combat simulation (randori), combat fundamentals learning (uchi-komi) and general athletic conditioning (running) to investigate their specific effect on internal training load, recovery state, physical enjoyment and well-being indices.”
Point 3: L 89: The number of male athletes in the present study differs from the other article using the same sample of participants, please correct.
Response 3: We apology for the mistake. Sentence was corrected and now starts as follows:
“Similar subject population and general methods were previously used for another study [10] and…”
Point 4: L 102-103: The authors indicate that athletes from the different experimental groups, but not control group, participated to a supplementary HIIT training in addition to their usual training. The use of “supplementary” would suggest that experimental groups performed an additional training, and that the HIIT training did not replace part of the usual training. Therefore, the total training volume would be lower for the control group compared to the experimental groups, which could influence perceived responses to training. Could the authors rephrase this sentence to avoid misleading whenever the HIIT training replace part of the conventional training, otherwise, authors should provide the total training volume for the experimental and control groups and account for this difference in their discussion.
Response 4: This strategy was used according to previous studies in combat sports that compared additional HIIT training compared with usual training program (Branco et al., 2017; Farzad et al., 2011; Franchini et al., 2016a; 2016b; Ravier et al., 2009). Namely, Branco et al. (2017) did not report any differences between experimental and control groups in terms of perceived recovery, rating of perceived exertion and internal training load using the same strategy of the present study.
Point 5: L 108 & 112: please provide the scales used to monitor RPE or DOMS, and details how DOMS were assessed and for which muscles groups.
Response 5: The scales used to monitor RPE and DOMS as well as details on how DOMS was assessed were added as follows:
“During these two periods, TQR scale from 6 to 20 (ranging from "very very poor recovery”, 6 points, to "very very good recovery”, 20 points) and well-being indices scores using a scale from 1 to 7 points (i.e., stress, sleep quality, fatigue level, DOMS and sum of these four variables used to calculate HI [12]) were collected 15 min before each training session. Delayed onset muscle soreness was used to rate the overall body soreness as previously reported by Papacosta et al. [9].”
Point 6: L 111: Could the authors provide references to support the methodology used to quantify session RPE? Indeed, an accurate monitoring of RPE should be conducted during exercise, or immediately at exercise completion (Pageaux B., 2016, European J of Sport Science)…
Response 6: We rigorously followed Foster et al. (2001) method (“Thirty minutes following the completion of each exercise bout, the subject was shown the RPE scale with verbal anchors (Figure 1) and asked to provide a rating of the overall difficulty of the exercise bout, the session RPE.” Foster, C.; Florhaug, J.A.; Franklin, J.; Gottschall, L.; Hrovatin, L.A.; Parker, S.; Doleshal, P.; Dodge, C. A new approach to monitoring exercise training. J Strength Cond Res 2001, 15, 109–115. doi:10.1519/00124278-200102000-00019. p. 111)
Point 7: The sole use of table in the results section is not well appropriate in this manuscript and somewhat boring to analyse. Authors should emphasize on main differences in the text, or the point of more interest regarding their analysis and discussion. I suggest therefore to reconsider the use of tables 1 and 2 by reducing the variables reported in the tables and clearly indicate in the text the most significant and relevant results.
Response 7: Most significant and relevant correlations were stated in 3. Results as follows:
“Most relevant correlations are the following. During intensified training period, TQR was negatively correlated with sleep (r = -0.5, p < 0.01, very large), stress (r = -0.35, p < 0.01, moderate), fatigue (r = -0.42, p < 0.01, moderate), DOMS (r = -0.35, p < 0.01, moderate), HI (r = -0.49, p < 0.01, moderate) and RPE (r = -0.62, p < 0.01, large). Moreover, RPE was positively correlated with sleep (r = 0.29, p < 0.05, low), fatigue (r = 0.40, p < 0.01, moderate), DOMS (r = 0.41, p < 0.01, moderate) and HI (r = 0.37, p < 0.01, moderate). Finally, physical enjoyment was negatively correlated with stress (r = -0.25, p < 0.05, low).
During tapering period, TQR was negatively correlated with sleep (r = -0.52, p < 0.01, large), stress (r = -0.68, p < 0.01, large), fatigue (r = -0.71, p < 0.01, very large), DOMS (r = -0.60, p < 0.01, large), HI (r = -0.77, p < 0.01, very large). Rating of perceived exertion was positively correlated with fatigue (r = 0.30, p < 0.05, moderate), DOMS (r = 0.26, p < 0.05, low).”
Former Tables 1 and 2 were changed to Tables S1 and S2 Supplementary Materials. Most significant and relevant results are discussed in 4. Discussion.
Point 8: L 195-197: In addition to the lower training volume, authors should also consider the psychological state of the athlete during a training period. Specifically, athletes may optimize themselves their positive feelings during this period, and coaches could also play a determinant role when conditioning athletes for competitions by focusing on their strengths and avoiding their weakness. How athletes perceived their recovery state may thus be biased and would therefore not relate accurately their physiological recovery level, which could partly support the low correlation between session RPE and DOMS mentioned L 201-202.
Response 8: We thank expert reviewer for his/her suggestion. It was embedded into the paragraph as study’s limitation as follows:
“Present study did not consider coaches’ role during tapering. Actually, coaches could influence athletes’ recovery perception, i.e., conditioning them to focus on their strengths and disregard their weaknesses. This could explain found low correlation between session-RPE and fatigue and DOMS.”
We hope that the manuscript has now reached the standard necessary for formal acceptance endorsement in International Journal of Environmental Research and Public Health.
We look forward to hearing from you.
Best regards
Reviewer 2 Report
In this study reported by Ouergui et al., relationships between several well-being indicators were investigated, either during intensified or tapering phases into a judo training program.
The study involves several analyzed variables, which were properly analyzed and supported by a well-conducted and concise statistical analysis.
Their results support that the analyzed well-being indices are negatively correlated during intensified and tapering periods, which could be influenced by the status of recovery and pre-fatigue periods in judo athletes.
The paper could be of utmost interest for high-performance athletes´ coaches.
Author Response
Response to Reviewer 2 Comments
Point 1: In this study reported by Ouergui et al., relationships between several well-being indicators were investigated, either during intensified or tapering phases into a judo training program.
The study involves several analysed variables, which were properly analysed and supported by a well-conducted and concise statistical analysis.
Their results support that the analysed well-being indices are negatively correlated during intensified and tapering periods, which could be influenced by the status of recovery and pre-fatigue periods in judo athletes.
The paper could be of utmost interest for high-performance athletes´ coaches.
Response 1: We thank expert reviewer for his/her positive comments.
We hope that the manuscript has now reached the standard necessary for formal acceptance endorsement in International Journal of Environmental Research and Public Health.
We look forward to hearing from you.
Best regards
Reviewer 3 Report
Subjective evaluation of the training load and the effects caused by this load is the basic feedback means of effective training management. From this point of view, the study is current. On the other hand, it contains several weaknesses that will need to be removed or supplemented. The abstract will need to be supplemented with more specific information about the monitored persons, at least age, sex, etc. At the same time, it is necessary to state the values ​​of correlation coefficients for the mentioned dependencies. Missing multiple regression information. At the end of the introduction, I recommend defining the working hypothesis and then discussing it in the relevant chapter. For multiple regression, the highest coefficient of determination is 38.7%, which other variables affect the monitored dependent variables. The reliability of the methods used is not defined, the data on monitoring, time, morning, morning, afternoon, period - preparatory, main are missing. The biological age of the observed adolescents is not described, only the chronological age is given, this should be discussed in the discussion chapter. The conclusion is vague, there is no specific procedure for using the results of the study for a particular athlete. The study will need to be substantially revised and re-evaluated.
Author Response
Response to Reviewer 3 Comments
Does the introduction provide sufficient background and include all relevant references?
(x) Must be improved
Please, read below comments to specific points.
Is the research design appropriate?
(x) Can be improved
Please, read below comments to specific points.
Are the methods adequately described?
(x) Can be improved
Please, read below comments to specific points.
Are the results clearly presented?
(x) Must be improved
Please, read below comments to specific points.
Are the conclusions supported by the results?
(x) Must be improved
Please, read below comments to specific points.
Point 1: The abstract will need to be supplemented with more specific information about the monitored persons, at least age, sex, etc.
Response 1: We thank expert reviewer for his/her suggestion. Specific information about investigated subjects (age, sex, etc.) were added to Abstract as follows:
“Sixty-one judo athletes (37 males, ranges 14-17 yrs, 159-172 cm, 51-67 kg) were randomly assigned to three experimental (i.e., randori, uchi-komi, running) and control groups (regular training).”
Point 2: At the same time, it is necessary to state the values of correlation coefficients for the mentioned dependencies.
Response 2: Values of correlation coefficients regarding investigated correlations are shown in Tables S1 and S2. This was further specified in 3. Results as follows:
“Results of correlational analyses (i.e., correlation coefficients and significances) between well-being indices, TQR, RPE and physical enjoyment measured over intensified and tapering periods for different judo training groups are presented in Tables S1 and S2, respectively.”
Point 3: Missing multiple regression information. … For multiple regression, the highest coefficient of determination is 38.7%, which other variables affect the monitored dependent variables?
Response 3: The highest coefficient of determination is not very high. Possible reasons that may explain this result were added in the text as follows:
“The highest coefficient of determination was 38.7% (Tables 1S and 2S) suggesting that only a small part of the variables could be explained by other variables. In the present study, this result may be associated with the fact that these relationships could be influenced by other variables such as physical fitness, training experience, sex, weight category and genetic endowment. Another possibility is that the constructs of each variable are quite independent and therefore could not be predicted by other variables.”
Point 4: At the end of the introduction, I recommend defining the working hypothesis and then discussing it in the relevant chapter.
Response 4: Aim’s paragraph was reworked as follows:
“To authors' current knowledge, studies in combat sports that have monitored internal training load and recovery state during extended intensified training followed by tapering periods are lacking. Over tapering periods, by definition, there is a training load decrease and this might variously influence internal training load, recovery state, physical enjoyment and well-being indices. Relationships between internal training load and recovery state have not been investigated in combat sports over a whole training period (i.e., intensified training+tapering). Thus, objective of the present study was to investigate the relationship between well-being indices (i.e., sleep, stress, fatigue, delayed onset muscle soreness [DOMS], Hooper index [HI]) and total recovery state [TQR]), internal training load (session-RPE) and physical enjoyment over intensified and tapering periods. It was hypothesized that physical enjoyment and internal training load are related to and predicted by well-being indices and recovery state.”
Point 5: The reliability of the methods used is not defined, …
Response 5: The reliability of methods used assessed in terms of Intraclass Correlation Coefficient was added as follows:
“… Well-being indices showed a good reliability in terms of Intraclass Correlation Coefficient (ICC) ranging from 0.60 to 0.81 in present study. … The TQR scale showed an ICC value of 0.823 in present study.
… Intraclass Correlation Coefficient for RPE resulted 0.808. … Physical Activity Enjoyment Scale showed an ICC value of 0.87 in present study. …”
Point 6: …the data on monitoring, time, morning, morning, afternoon, period - preparatory, main are missing.
Response 6: Data on training sessions and measures time of day and period were added to 2. Materials and Methods and now requested information says as follows:
“After being well familiarized with all scales and questionnaires, in addition to their usual technical-tactical training, athletes participated in supplementary HIIT – except participants in the control group – over 4 weeks with two sessions per week followed by twelve days of tapering (time). …
During these two periods, TQR scale from 6 to 20 (ranging from "very very poor recovery”, 6 points, to "very very good recovery”, 20 points) and well-being indices scores using a scale from 1 to 7 points (i.e., stress, sleep quality, fatigue level, DOMS and sum of these four variables used to calculate HI [12]) were collected 15 min before each training session (monitoring). …
Rating of perceived exertion (RPE, Borg’s CR-10 scale [14]) was recorded for each athlete 15-30 min after each training session and perceived training load (i.e., session-RPE) following each session for each judo athlete was calculated from the product of the session duration and the athlete’s RPE [14] (monitoring). … Moreover, physical enjoyment calculated from the sum of 18 items of the Physical Activity Enjoyment Scale (PACES [15]) was assessed (monitoring) after both intensified and tapering training periods (intensified and tapering). All training sessions and measures were conducted at same time of day (5:00 pm-7:00 pm [16]) during the competition period of the season.”
Point 7: The biological age of the observed adolescents is not described, only the chronological age is given, this should be discussed in the discussion chapter.
Response 7: Following limitation paragraph was added to 4. Discussion:
“A limitation of present study is that biological age of participants was not controlled. Despite being similar in chronological age (14-17 yrs), they were still in the process of growth and maturation and that could have influenced their internal training load, recovery state, physical enjoyment and well-being indices measures.”
Point 8: The conclusion is vague, there is no specific procedure for using the results of the study for a particular athlete.
Response 8: We thank expert reviewer for his/her suggestion. Conclusion was made clearer by detailing last sentence as follows:
“Thus, coaches can use simple and low-cost monitoring tools such as the well-being indices sleep and fatigue (particularly before training), total recovery state (TQR) scale and Hooper index (HI) during both intensified and tapering periods to control training stimuli and to adjust training regimens when necessary to support positive psychological states and properly-timed physiological recovery among judo athletes.”
We hope that the manuscript has now reached the standard necessary for formal acceptance endorsement in International Journal of Environmental Research and Public Health.
We look forward to hearing from you.
Best regards
Round 2
Reviewer 3 Report
The text has been edited and supplemented according to the comments of the opponent. As a rule, the results were added. A certain margin is a conclusion that is increasingly descriptive than specifically usable data. On the other hand, in this form, the study can be recommended for publication.